# Customer Satisfaction with Farmhouse Facilities and Its Implications for the Promotion of Agritourism Resources in Italian Municipalities

**Rosa Maria Fanelli \*** and **Luca Romagnoli**

Department of Economics, University of Molise, 86100 Campobasso, Italy; luca.romagnoli@unimol.it
* Correspondence: rfanelli@unimol.it; Tel.: +39-874-404401

**Abstract:** The importance of a website as a distribution and promotional channel in the context of the tourism sector is increasingly relevant. Despite this, the availability of online reviews referring to agritourism resources is very rarely addressed in academic literature. This study investigates visitor satisfaction with farmhouses on the basis of online reviews available on the official websites of 397 farmhouses with educational farms associated with the Agritourism.it organization. Visitor satisfaction is important for successful destination marketing as it influences destination selection and the consumption of products and services. This study's findings provide insights for consumers and agritourism operators with the aim of matching customer expectations to the types of agritourism on offer in Italian municipalities. On the demand side, according to the overall satisfaction with the specific characteristics of farmhouses as expressed by 10,864 visitors, the results of a Principal Component Analysis showed that the activities and facilities present in the surrounding natural and cultural areas of the farmhouses are the attributes with the highest visitor satisfaction scores. In contrast, on the supply side, the results of a Hierarchical Cluster Analysis showed different groups of farmhouses characterized by "homogenous" features and by a "homogenous" heritage of natural and cultural resources. Assessing visitor satisfaction and feedback can help managers to improve their service performance. For this purpose, the findings of the study reveal that the economic and environmental sustainability, among the other motivations, has shown to play a crucial role in influencing visitors' frequency to learn about specific information concerning agriculture, the environment, and about issues regarding food, culture and the rural tradition.

**Keywords:** farmhouse; resources; Hierarchical Cluster Analysis; Italian municipalities; online review; Principal Component Analysis; visitor satisfaction

## 1. Introduction

### 1.1. Customer Satisfaction and Online Reviews

In the tourism sector, several studies have explored customer satisfaction and have highlighted how it plays a fundamental role in encouraging visitors to return to a location, to make recommendations to their friends, and to give positive and negative reviews, thus affecting customer behavioral loyalty [1–9]. In this context, word of mouth (WOM) is one of the most influential factors affecting consumer behaviour [10]. This influence is especially important with intangible products that are difficult to evaluate prior to consumption, such as tourism or hospitality. Indeed, the hospitality sector is characterised by customers who are heterogeneous in terms of expectations, needs and service perceptions, which are crucial for understanding customer differences. The consequence is that in recent years, online reviews have transformed consumer behaviour in terms of information

searching and sharing. Indeed, in the literature, there are indications that online consumer reviews have a significant influence on travel information searches and product sales [11–14]. Online review websites allow consumers to search for detailed and reliable information by sharing their consumption experiences [15,16] with the aim of reducing their level of perceived uncertainty [17]. Therefore, on the demand side, visitors, also known as customers, consumers or buyers of any sector of tourism, are one of the most important components of tourism management. Indeed, visitor satisfaction is important for successful destination marketing as it influences the selection of the destination, the consumption of products and services, publicity WOM, and the decision to return [18]. In some studies [19], the satisfaction of tourists has been measured by general satisfaction attributes (e.g., attractions, accommodation, accessibility, amenities and activities) and has been found to meet expectations. In contrast, in terms of offer, it represents a rich and useful source for customers and marketers evaluating product and service quality [20]. In comparison to traditional survey methods, user generated content (UGC) is considered more objective, vast and free from sample bias, in that posts are made spontaneously and are devoid of laboratory effects [21,22]. In addition, UGC is inexpensive and relatively easy to collect [23].

Despite the growing body of online evaluation research in the tourism sector, in the agritourism sector, the literature remains inconclusive regarding the evaluation of farmhouse attributes, especially when concerning the judgments made by visitors. The aim of this study was to fill this gap and to contribute to a better understanding of the relationships between farmhouse features and visitor-expressed satisfaction with the farmhouse attributes and with the heritage of natural and cultural resources present in the surrounding area where the facility is localised.

### 1.2. Farmhouse Facilities with an Educational Farm

The concept of agritourism has been discussed in a variety of contexts in the international literature relating to tourism, rural development and visitor experience [24–27]. Agritourism can be considered an activity that links the economic, social and environmental components of sustainability [28]. In rural sustainable development, with specific effects on the environment, agricultural heritage, gastronomic and economic growth, an important role can be played by the farmhouse facilities with an educational farm (and referred to herein as "FFEFs, or only "Farmhouses"). In these structures, the farmer and their family members organise educational, recreational and leisure activities for visitors (e.g., hosts children, youth, school trips, as well as other groups and private individuals) as part of their normal work [29]. Furthermore, educational farms, that serve a number of social and economic functions, should be considered as a business model oriented toward sustainability and can support very well the sustainable development of rural areas [30,31]. Indeed, in these areas, FFEFs realise important synergies among agriculture, gastronomy, territory and tourism [32–35]. Instead, from a visitor perspective, educational farms can be a place where the experiential approach [36] finds a fruitful sphere of application from the demand side [37]. However, visitors can gather experiences with plants and animals, and those which illustrate the meaning of sustainable and product-oriented production of food.

In Poland, for example, educational farms, as approved by the Ministry of Agriculture and Rural Development, are defined as "the activity conducted by rural inhabitants and located in rural areas." This activity is involved in the implementation of at least two of the following objectives: education in terms of plant and/or animal production and the processing of agricultural products; education in terms of environmental and consumer awareness; education in terms of the material heritage of rural areas in handicraft, folk art and traditional occupations [38].

In this direction, educational tourism in agriculture has been gaining popularity in many countries as an emerging potential market segment of rural tourism, along with growing demands for experience-oriented tourism [39]. Indeed, the very first farms, where city-dwellers could go and learn about rural life, were born in Scandinavia, in Norway, Denmark and Sweden. Gradually, the idea of creating farms for this kind of activity travelled south, reaching the Mediterranean during the 1970s.

In Italy, the first real project of this type was established in 1967 thanks to Alimos, a society of agronomists and agricultural technicians who wanted to encourage innovation in the fruit and vegetable sector, to support environmental protection and to spread a new food culture [40]. These structures have had an important role in education policies regarding food consumption and nutrition [29]. However, the FFEFs were created mainly to supplement income in an agricultural world that was struggling with low farm budgets and to encourage young farmers in the sector. Therefore, educational farms are a good example of enterprises with multi-functionality, as defined by the Organisation for Economic Co-operation and Development (OECD) [41] and of a location in which tourists can satisfy their desire to reconnect with the cultural roots of food and to participate in the typical educational activities of a working farm [42,43]. The FFEFs usually offer apartments equipped with independent facilities and often offer breakfast or half/full board that involves serving seasonal products of their own production or from local farms. Most of the accommodation is located in original structures, renovated to preserve their original architectural forms or modernised with swimming pools. They often have thick stone walls, exposed wooden beams, and terracotta roofs and floors. Easy access is an important aspect in FFEF and most tourists expect it [44]. Indeed, visitors prefer convenient and accessible locations [45–47]. Visitors do not expect, however, high quality accommodation facilities in an urban style; they expect country accommodation facilities which are well maintained, clean and pollution free [48,49]. Such accommodation may be farmhouses, cottages, castles, masserie (large stone farmhouses) and so on [44]. Furthermore, landscape attributes appear to play a more significant role in agritourism consumer preferences for overnight destinations [50–54]. However, the success of FFEFs in Tuscany is mainly due to the beautiful landscape, the high-quality food products and the many historical centres spread out in the neighbouring countryside (e.g., San Gimignano, Volterra, Cortona in Tuscany) or nearby (Florence, Pisa, Lucca). The experience of FFEFs, as reported in several recent studies [55–59], is an increasingly worldwide phenomenon, which has also seen growth in all Italian regions, especially in Tuscany, Umbria, Sicily and Apulia. In these regions, there has been a significant diffusion of farmhouses specialising in offering overnight accommodation, education and social services in rural areas [60]. Furthermore, some scholars [61] have highlighted that Italian FFEFs are generally considered to be an alternative and cheaper form of overnight accommodation, where tourists can spend their holidays staying close to the more well-known tourist destinations (Rome, Florence, Venice), or to regions such as Tuscany, Umbria, Sicily and Apulia that belong to specific areas of tourism or rural and agri-food districts. Indeed, farmhouses and certified quality agri-food products have helped to lead the Italian primary sector towards consolidated development in certain rural areas.

As mentioned above, in Italy, the FFEF remains a niche sector that can only be offered by a farmer and their family members (Law n. 96/2006). However, the absence of a national regulatory framework is reflected in the lack of a database on the sector and in the uncertainty about the dimension of the phenomenon. As of 2017, the number of farms registered in the regional lists of educational farms is equal to 2291, but the phenomenon could be larger given the failure to establish some regional registers. According to some estimates [62], the global number of farms in Italy is around 2500-2900 that carry out educational activities on an ongoing basis, of which about 60% are made up of agritourism businesses. Finally, based on ISTAT data (2017) [63], around 1500 farms also carry out educational activities and the phenomenon seems to be growing strongly.

In light of the above and to fill the gap in the literature concerning the relationships between farmhouse features and visitor satisfaction, the aim of this study was to investigate visitor satisfaction with and evaluation of the characteristics and attributes of farmhouses in Italian municipalities on the basis of reviews extrapolated from the official websites of Italian farmhouses. The starting point for the research were studies by Santeramo and Barbieri (2016) [47] and Phillip et al. (2010) [64], which clearly state the need to identify the characteristics of different types of accommodation offered by farmhouses as well as their different attributes, in order to match them to the different expectations of visitors. In this way, business models outside traditional spheres can be developed.

To enrich this aim, the following research questions were set:

**RQ1.** What types of FFEFs are present in Italian municipalities?
**RQ2.** How did visitors evaluate their stays in FFEFs?
**RQ3.** Which forms of FFEFs are best suited to which types of visitors?
**RQ4.** How do different online visitor judgments contribute to promoting natural and cultural resources present within the different contexts of the Italian agritourism sector?

## 2. Materials and Methods

### 2.1. Data Collection

The information for this study was collected from the website of the Italian Association FFEF [65], which consisted of 631 members in April, 2019. Agriturismo.it is the reference point both for tourists looking for a farm and for the people who manage them. In addition, it is the only site that solely promotes real farmhouses that are authorised *agriturismi* and not merely B&Bs or holiday country houses. The site offers accurate descriptions and useful tools to easily find the most suitable farm for everyone. Furthermore, Agriturismo.it is part of the Feries family, the Italian leader in online extra-hotel accommodation that also manages CaseVacanza.it. The site was created in 2001 and in 18 years, the managing of family holidays has developed many different skills. The current study involves a dataset based on 10,864 visitor reviews extrapolated from the websites of 397 of the 631 of Italian FFEF (about 63%) associated with this site, which operate in 341 municipalities (4.3% of the Italian municipalities) of the 20 Italian regions. The remaining 36% were not analysed because they did not possess all the information necessary for the analysis. On average, there were 27.4 reviews per farmhouse and about 32 reviews per municipality. Over 30% of the farmhouses received less than ten reviews. In contrast, about 23% of all the reviews were obtained by the top 17 farmhouses. The websites provided information both on the offer side (farmhouse dimensions, prices and the services offered by the farmhouses associated with the Agriturismo.it organisation) and the demand side (visitor evaluations of the principal farmhouse attributes). As is well-known, a website has a large market share, operating on a commission-based model with a large and active community, which continually generates valuable feedback and information. In addition, it provides a distinction between positive and negative parts of reviews, an aspect that was important for the types of analysis carried out. As shown in Table 1, the majority of FFEFs considered in this study (42.6%) are located in Central Italy, followed by 23.3% located in Northern Italy, 17.3% in Southern Italy and 16.8% in the islands of Sicily and Sardinia. The first four regions with the highest number of farmhouses associated with the Agriturismo.it organisation (more than 40) are Tuscany (116, 18.4% on the total), which is considered to be the leading Italian region for rural tourism, Umbria (51, 8.1%), Sicily (50, 7.9%) and Apulia (44, 7%). Tuscany is "the" hot travel destination in Italy, especially for food and landscape lovers. More than half of FFEFs (55%) are situated in low-lying areas, 32% in hilly areas and 13% in mountainous areas. This may be due to the specific features of FFEF: they can better operate and offer their services in low-lying areas.

On the basis of this list, from March to April 2019, the names and locations of all the FFEFs in each municipality were downloaded. The variables extrapolated from the websites were 15 (Table 2), but for the analysis they were reduced to 11: the number of services (NS); the number of beds (NB); the number of accommodation units (NA); the number of beds in large accommodation (NBLA); the price per night for one guest (PNOG); the ratio between the number of reviews (NR) and the number of years (NY) during which the visitors gave their evaluation, which is considered as a proxy for the consolidated farmhouse websites (NR.NY); the average reviewer judgment of all the characteristics (SCOREALL); and the average reviewer judgment of the most important attributes of the farmhouses: settings and surroundings (SCORESETSUR); activities and facilities (SCOREACTFAC); accommodation (SCOREACCOM) and restaurants (SCOREREST).

**Table 1.** Farmhouse Facilities with an Educational Farm (FFEFs).

| Region | (A) Total no. of FFEFs | (B) No. of FFEFs Considered | (C) No. of Municipalities | B/A |
|---|---|---|---|---|
| Tuscany | 116 | 62 | 50 | 53.45 |
| Umbria | 51 | 33 | 23 | 64.71 |
| Sicily | 50 | 35 | 29 | 70.00 |
| Apulia | 44 | 31 | 27 | 70.45 |
| Sardinia | 36 | 32 | 27 | 88.89 |
| Campania | 34 | 24 | 20 | 70.59 |
| Liguria | 34 | 20 | 19 | 58.82 |
| Lombardy | 33 | 22 | 22 | 66.67 |
| Veneto | 29 | 16 | 16 | 55.17 |
| Trentino Alto Adige | 29 | 11 | 9 | 37.93 |
| Lazio | 29 | 18 | 14 | 62.07 |
| Emilia Romagna | 26 | 13 | 12 | 50.00 |
| Piedmont | 26 | 19 | 16 | 73.08 |
| Marche | 26 | 16 | 15 | 61.54 |
| Abruzzo | 25 | 20 | 18 | 80.00 |
| Calabria | 18 | 10 | 9 | 55.56 |
| Friuli Venezia Giulia | 12 | 7 | 7 | 58.33 |
| Basilicata | 8 | 5 | 5 | 62.50 |
| Molise | 3 | 2 | 2 | 66.67 |
| Aosta Valley | 2 | 1 | 1 | 50.00 |
| **Total** | **631** | **397** | **341** | **62.92** |

Source: Authors' elaboration of data from Agritourism.it, 2018.

**Table 2.** List of starting variables.

| Variables | Coding | Description |
|---|---|---|
| Altitude | ALTITUDE | Altitude of the municipality where the farmhouse is located (metres above sea level (ASL)) |
| Wi-Fi | WI.FI | Equipment for Wi-Fi connectivity |
| Number of services | NS | Number of services offered by farmhouse |
| Number of beds | NB | Number of beds in farmhouse |
| Number of accommodation units | NA | Number of accommodation units in farmhouse |
| Number of beds in largest accommodation | NBLA | Number of beds in large accommodation in farmhouse |
| Price night/one guest | PNOG | Price per guest for one night in farmhouse (€) |
| Number of reviews | NR | Number of reviews posted on farmhouse website |
| Score for all characteristics | SCOREALL | Reviewer judgements of all farmhouse attributes |
| Score for setting and surroundings | SCORESETSUR | Reviewer judgements of farmhouse surrounding area and context |
| Score for activities and facilities | SCOREACTFAC | Reviewer judgements of farmhouse activities and facilities |
| Score for accommodation | SCOREACCOM | Reviewer judgements of farmhouse accommodation |
| Score restaurant | SCOREREST | Reviewer judgements of restaurant (foods and beverages) |
| Number of years for the judgement | NY | Number of years of farmhouse evaluation |
| Number reviews/number of years | NR.NY | Farmhouse consolidated website (proxy) |

Source: Authors' elaboration of data from Agritourism.it, 2018.

## 2.2. Data Analysis

Many research studies have assessed demand in the agritourism sector using quantitative methodologies, which can be broadly subdivided into two main categories: stated preference methods and revealed methods [47]. The most important quantitative methods employed in the existing

literature have focused on the gravity model, on the travel cost method, and on the hedonic regression model [47]. All of these methods fall into a modelling framework, that is, they intend to explain the agritourism demand (most often considered the dependent variable) by means of the so-called "drivers"—the (independent) variables influencing tourist choices.

In this research, a different standpoint has been assumed: some demand drivers, together with some variables referring to the appreciation of the tourists, have been extrapolated from the FFEF websites, not with the aim to embed them into a model of tourism demand, but to characterise the types of farmhouses both from the demand and of the offer point of view. The approach followed was that of exploratory multivariate methods [66], which were applied on the 11 indicators extrapolated.

In particular, statistical analysis was performed in 4 steps: (a) descriptive statistics, (b) principal component analysis; (c) hierarchical cluster analysis, followed by (d) a visual mapping of the spatial distribution of the clusters resulting in step (c). All analyses were carried out by means of R statistical software (version 3.5.3).

(a) *Descriptive analysis* allows the conversion of raw data into a form that would make them easy to understand and interpret through rearranging, ordering, and manipulating data to generate descriptive information [67]; descriptive analysis was applied to describe the different features and attributes of the farmhouse (Table 2);

(b) *Principal component analysis (PCA)* was employed as a dimensionality reduction tool [68]. It allowed the number of quantitative variables to be reduced into a smaller set of factors or principle components (PCs). Since the starting variables were characterised by different units of measurements, we performed PCA on the standardised values:

$$z_{ij} = \frac{x_{ij} - \mu_{j(X)}}{\sigma_{j(X)}}$$

where $x_{ij}$ is the generic entry of the data matrix $X$, with $n$ rows (farms) and $p$ columns (variables); $\mu_{j(X)}$ and $\sigma_{j(X)}$ are the mean and standard deviation of the $j$-th variable, respectively, and $z_{ij}$ is the (dimensionless) entry of the standardised matrix $Z$, having $\mu_{j(Z)} = 0$ and $\sigma_{j(Z)} = 1$.

(c) *Hierarchical cluster analysis (HCA)*. After performing PCA, HCA based on Euclidean distances was conducted on the first 4 PCs, explaining a relevant part of original cumulative variance. Ward's classification algorithm was used. This method is distinct from all others since it uses an analysis of variance approach to evaluate the distances between clusters. We can refer to Ward (1963) [69] for details concerning this method, which is regarded as very efficient; it tries to find the partitions $P_n, P_{n-1}, \ldots, P_1$ in a manner that minimises the loss associated with each grouping, and to quantify that loss in a form that is readily interpretable. The Ward method relies upon the well-known decomposition: $T = W + B$, where $T$ is the total sum of squares (SS) of the observations, $W$ is the within-clusters SS, and $B$ is the between-clusters SS. In general, passing from $k + 1$ to $k$ clusters, $W$ tends to increase (less homogeneity in the new cluster with the addition of new units), while of course $B$ decreases: at each step of the Ward procedure, the clusters joined together are the two with the minimum increase in $W$.

Regarding the subsequent choice of an optimal number of clusters, it is common practice to repeat the analysis for different numbers of clusters, and then calculate the objective function.

$$R^2_{(k)} = \frac{B_{(k)}}{T}$$

This procedure is sometimes called the *elbow method*, since, when reporting on a graph the number of clusters $k$ (on the horizontal axis) and the $R^2_{(k)}$ values (on the vertical axis), a good choice for the number of clusters is the $k$ where the graph presents an "elbow", i.e., a sudden change in slope.

(d) *Visual mapping of the clusters and centrographic analysis.* Following O'Sullivan and Unwin (2010) [70], simple descriptive centrographic measures are presented, to provide summary descriptions

of the point patterns of the clusters resulting from step (c). In particular, the *mean centres* and the *standard distances* of the clusters are shown, where

$$c_j = \left( \mu_{j(X)}, \mu_{j(Y)} \right) = \left( \frac{\sum_{i=1}^{n_j} x_{ij}}{n_j}, \frac{\sum_{i=1}^{n_j} y_{ij}}{n_j} \right)$$

is the mean centre of the *j*-th cluster, whose coordinates, $\mu_{j(X)}$ and $\mu_{j(Y)}$, are the means of the coordinates (longitude and latitude) of the $n_j$ points (farmhouse) in cluster *j*, with $j = 1, \ldots, 8$, and

$$d_j = \sqrt{\frac{\sum_{i=1}^{n_j} \left[ \left( x_{ij} - \mu_{j(X)} \right)^2 + \left( y_{ij} - \mu_{j(Y)} \right)^2 \right]}{n_j}}$$

is the standard distance for cluster *j*; it is strictly correlated to the statistical concept of standard deviation and, in this framework, it expresses how the farmhouse belonging to the *j*-th cluster are dispersed around their centre of gravity, $c_j$.

## 3. Results

### 3.1. The Features of Italian Farmhouses and Visitor Evaluations

The first question of this study aimed to identify and analyse the different typologies of FFEFs present within the natural and cultural contexts of Italian municipalities.

A summary of the features of these farmhouses are reported in Table 3. Farmhouses are predominantly located in hilly areas with an average altitude of 314 metres ASL and, on the basis of the NB, NA and NBLA, are of diverse dimensions. The average NB and NA are equal to 22 and 7 respectively. Regarding the variable NBLA, the coefficient of variation (CV) highlights the biggest difference (84.84%) among the 397 farmhouses considered in the analysis. Another characteristic of the farmhouses is that, on average, they have about eight services (7.64). The value of CV (33.26%) indicates that there are quite a number of differences between the farmhouses in relation to the number of services available. Examples of the services on offer are: Internet, air conditioning, swimming pools, wellness centres, playgrounds for children, facilities for pets, barbecues, restaurants, educational opportunities (e.g., cookery classes, cannery tours and wine tasting), on-site direct sale of typical farm products (especially cheeses, oil and wine), disabled facilities, credit card facilities, attractions in the surrounding area (e.g., nature reserves, museums, galleries and so on), outdoor activities (e.g., hiking, bird watching, hunting, fishing, horse riding, diving, skiing, tennis). Furthermore, diversification on farms provides visitors with educational opportunities, such as learning about the role of the agricultural sector in the community and local economy. Another value of CV close to 84% is recorded for the NR.NY, which is considered as a proxy for a website that brings together customer reviews and services. This indicates that FFEFs with a consolidated presence of websites coexist in the agritourism sector with those that have only more recently introduced this important tool.

In relation to the PNOG, the average value is about €37 and there is quite a variation between the 397 farmhouses (36.07). Despite this dissimilarity, there is greater homogeneity between the same farmhouses regarding the visitor evaluation of farmhouse attributes. This confirms the above results that farmhouses try to offer a wide range of services to satisfy, in a better way, visitor expectations.

In relation to the second research question: *How did visitors evaluate their stays in FFEFs?* It seems that visitors, on average, were more satisfied with the restaurants (9.43) and less so with activities and facilities in the natural and cultural surrounding areas (9.06).

**Table 3.** Descriptive analysis of the farmhouse characteristics and of the visitor evaluations.

| Variable | Mean | Std. Dev. | C.V. | Median | 1st Qu. | 3rd Qu. | Min | Max | Skewness | Kurtosis |
|---|---|---|---|---|---|---|---|---|---|---|
| NB | 22.16 | 13.24 | 59.74 | 19.00 | 13.00 | 27.00 | 2.00 | 73.00 | 1.47 | 2.46 |
| NA. | 7.30 | 5.07 | 69.45 | 6.00 | 4.00 | 9.00 | 1.00 | 48.00 | 2.92 | 14.66 |
| NBLA | 5.66 | 4.80 | 84.84 | 5.66 | 4.00 | 6.00 | 2.00 | 60.00 | 6.80 | 61.58 |
| PNOG | 36.79 | 13.27 | 36.07 | 35.00 | 30.00 | 40.00 | 7.50 | 125.00 | 2.35 | 10.45 |
| SCOREALL | 9.41 | 0.64 | 6.76 | 9.60 | 9.20 | 9.60 | 5.5 | 10.0 | -2.46 | 9.02 |
| SCORESETSUR | 9.20 | 0.66 | 7.22 | 9.30 | 8.90 | 9.60 | 5.5 | 10.0 | -1.68 | 4.56 |
| SCOREACTFAC | 9.06 | 0.73 | 8.00 | 9.40 | 8.70 | 9.60 | 5.5 | 10.0 | -1.34 | 2.85 |
| SCOREACCOM | 9.24 | 0.70 | 7.58 | 9.40 | 8.90 | 9.80 | 5.5 | 10.0 | -1.59 | 3.95 |
| SCOREREST | 9.43 | 0.70 | 7.45 | 9.60 | 9.20 | 9.90 | 4.3 | 10.0 | -2.82 | 12.14 |
| NR.NY | 3.85 | 3.22 | 83.42 | 2.83 | 1.67 | 5.00 | 0.40 | 21.15 | 1.95 | 4.80 |
| NS_WI.FI | 7.64 | 2.54 | 33.26 | 7.00 | 6.00 | 9.00 | 1.00 | 13.00 | 0.42 | -0.16 |

Source: Own work based on collected materials. The dataset of the 11 variables employed in the study could be found in the Supplementary Materials.

The results of the correlation analysis (Figure 1), show that the overall evaluation from visitors (SCOREALL) of farmhouse attributes was generally, positively correlated with SCOREACTFAC (0.72), with SCOREREST (0.68), with SCOREACCOM (0.67) and less so with SCORESETSUR (0.57). NB is linked positively with NA (0.6). In contrast, NB and NA show a low correlation with NS_Wi.Fi, equal to 0.24 and 0.25, respectively. These correlations shed some light on the possible links between the original variables from which the PCs derived, which are likely to show a subsequent merging of those variables showing stronger correlations. Another relevant observation is that, on the one hand, the negative relationship between the PNGO and the presence of a website with customer review data suggests that if the PNGO increases, the number of written reviews decreases. On the other hand, it is clear that the NS exerts little influence on the PNGO. Consequently, NS is not considered as important as the environmental and landscape features of the surrounding areas, so that the costs of maintaining the aesthetic qualities of the landscape probably are internalised in the price of staying in the FFEF.

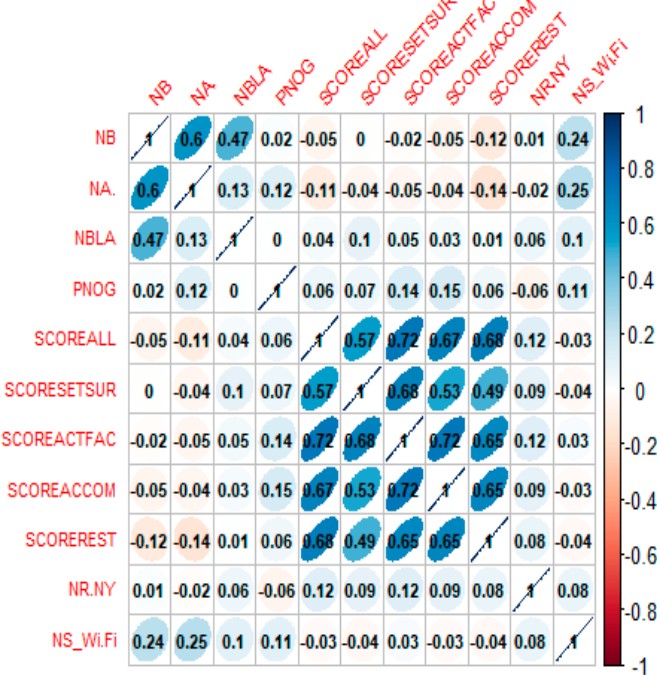

**Figure 1.** Pairwise correlation matrix among starting variables.

## 3.2. PCA Application to Highlight Latent Dimensions in the FFEF Data

The PCA extracted four PCs with eigenvalues ≥1, explaining a cumulative percentage of 70% of the total variance in the (standardised) dataset. The first component accounted for 32.82% of the total variance, the second for 18.03%, the third for 10.13% and the fourth for 9.09% (Table 4).

**Table 4.** Eigenvalues analysis.

| Component | Eigenvalue | % of Total Variance | Cumulative % of Total Variance |
|:---:|:---:|:---:|:---:|
| 1 | 3.610 | 32.822 | 32.822 |
| 2 | 1.983 | 18.027 | 50.849 |
| 3 | 1.114 | 10.126 | 60.975 |
| 4 | 1.000 | 9.087 | 70.063 |
| 5 | 0.822 | 7.472 | 77.540 |
| 6 | 0.766 | 6.963 | 84.498 |
| 7 | 0.545 | 4.959 | 89.457 |
| 8 | 0.343 | 3.115 | 92.572 |
| 9 | 0.318 | 2.889 | 95.460 |
| 10 | 0.278 | 2.523 | 97.983 |
| 11 | 0.222 | 2.017 | 100.00 |

Source: Own work based on collected materials.

The meaning of each PC is determined by the original variables correlated with it. The correlations indicate high loadings of five variables on component 1 (PC1), and all of them concern the visitor evaluation scores. As a result, it is possible to define PC1 as a synthesis of the "*visitor evaluation scores*" of the farmhouse—as could be expected by examining the pairwise correlation matrix, see Table 4.

For its part, PC2 represents the "*dimensionality attributes*" of farmhouses, with high loadings on the NB, NA and NBLA. This result indicates that the physical dimensions of farmhouses are an important factor, since they influence both the process of productive diversification into agritourism activities and the efficiency of the farmhouses themselves.

Component 3, the "*value factor*", has a high correlation (0.76) with the PNOG. This indicates that satisfaction is also affected by price. Therefore, any attempt to measure satisfaction needs to take the value factor into account.

Component 4 has high loadings on the number of reviews/number of years during which the visitors made their judgments (NR.NY) and on the number of services (NS_WI.FI), so that it could be defined as a summary expression of the "*consolidated website and services farmhouse*" (Figure 2). In this latter case, farmhouse websites are very important in promoting the landscape and the services offered, especially in the case of small dimension FFEFs located in distant rural areas.

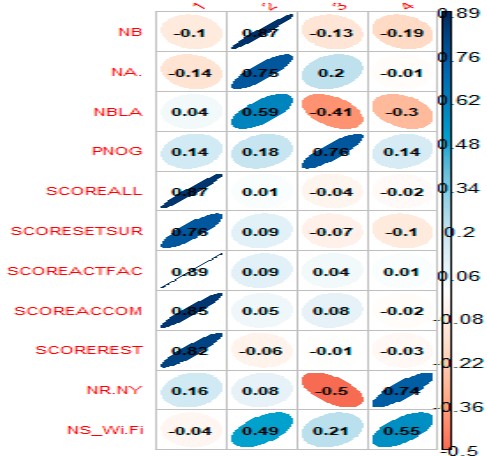

**Figure 2.** Correlations among starting variables and PCs.

### 3.3. Classification of Italian Farmhouses into "Homogenous" Groups

In the second step of the analysis, HCA was adopted to identify which forms of farmhouses were suited to which types of visitors (RQ$_3$). Based on the 4 PCs extracted in the previous step—PC1: the "*visitor evaluation scores*"; PC2: the "*dimensionality attributes*"; PC3: the "*value factor*"; PC4: the "*consolidated website and services farmhouse*"—a HCA with Ward's method (see Section 3.2) was carried out too. The choice of the optimal number of clusters followed the "elbow method", and Figure 3a shows an elbow at the 8 cluster solution, with the resulting subdivision accounting for 62% of the total variance. Figure 3b reports the percentage composition of the cluster numerosity.

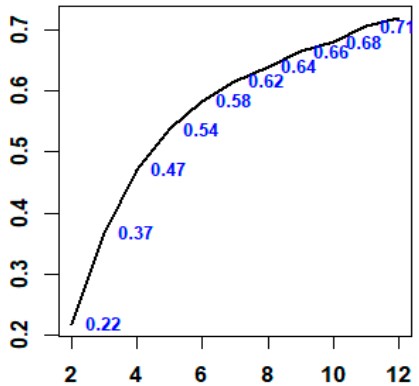 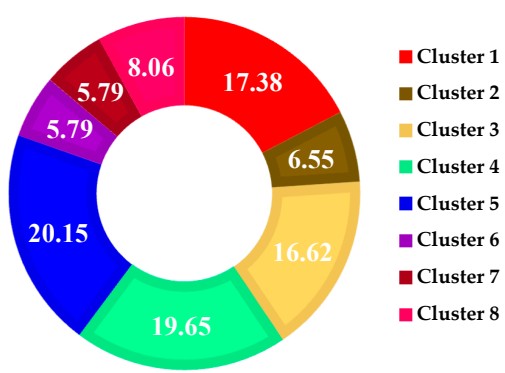

**(a)** Choice of optimal clustering solution: number of clusters, $k$ (horizontal axis) plotted versus $R^2_{(k)}$ (vertical axis)

**(b)** Pie chart showing the relative clusters sizes (percentage of the total number of farmhouse per each cluster)

**Figure 3.** Clustering summary.

In this section, the main features of the eight clusters will be highlighted, firstly on the basis of their geographical distribution (Section 3.3.1).

#### 3.3.1. Mapping a HCA Solution

The results of the HCA are visually displayed in Figures 4 and 5. The first shows the territorial distribution of the farmhouses into clusters, and the second presents the geographic location of the mean cluster centres.

Figure 5 clearly shows that each of the clusters does not possess any kind of spatial clustering: all 8 cartograms consist of very dispersed points (farmhouses), which always belong to all of the territorial subdivisions: Northern, Central and Southern Italy.

As can be seen in Figure 5, the mean centres lie in a restricted area around Central Italy: this means that the clusters have similar centres of gravity. The second centrographic measure, the standard distance of the farmhouses from their mean centre, is reported in the following.

The visual inspection of Figure 5 finds confirmation in Table 5: all the standard distances are very large, only cluster 8 is a little below the 300km threshold; the clusters consist of very dispersed farmhouses. The final consideration for this subsection is that the characterisation of Italian farmhouses does not involve geographic neighbourhood features, but only a detailed study of the farmhouses, independent of their spatial locations.

**Table 5.** Standard distances for the 8 clusters (in km).

| Cluster 1 | Cluster 2 | Cluster 3 | Cluster 4 | Cluster 5 | Cluster 6 | Cluster 7 | Cluster 8 |
|-----------|-----------|-----------|-----------|-----------|-----------|-----------|-----------|
| 374.5 | 341.1 | 337.6 | 351.4 | 323.7 | 387.5 | 330.9 | 293.6 |

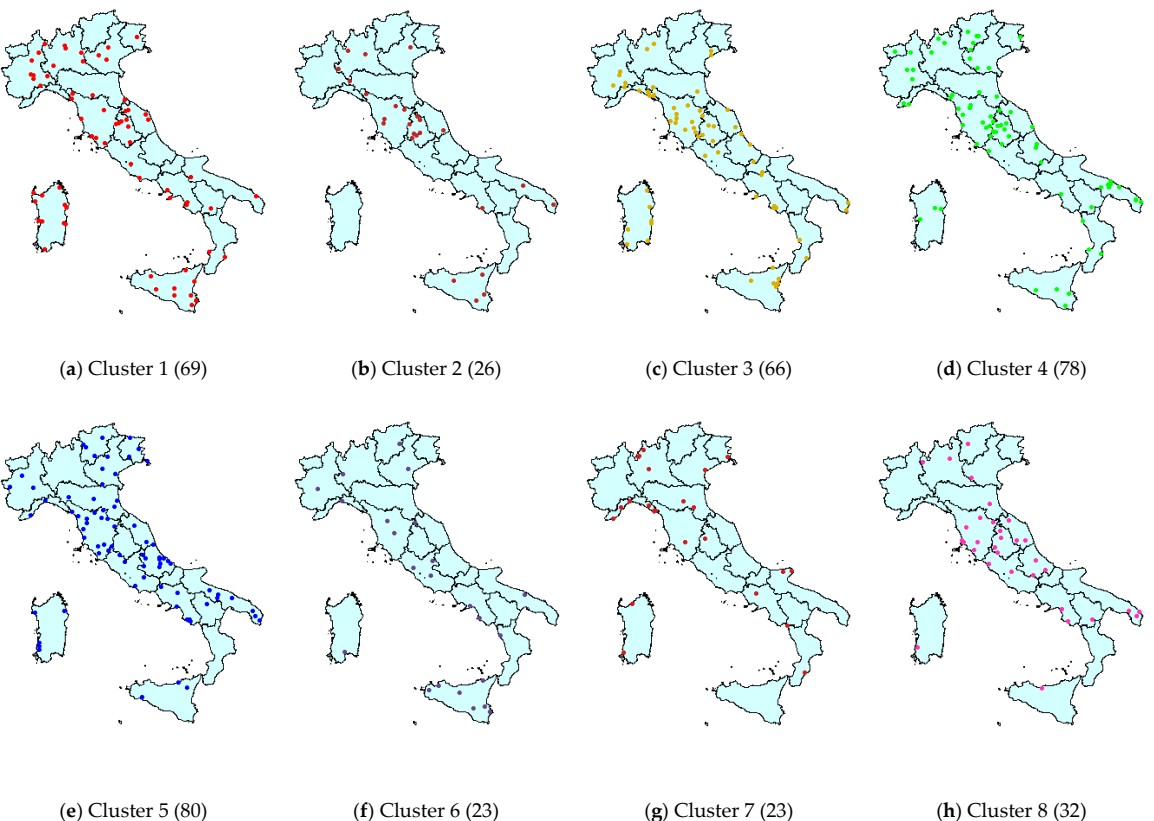

(**a**) Cluster 1 (69)　　(**b**) Cluster 2 (26)　　(**c**) Cluster 3 (66)　　(**d**) Cluster 4 (78)

(**e**) Cluster 5 (80)　　(**f**) Cluster 6 (23)　　(**g**) Cluster 7 (23)　　(**h**) Cluster 8 (32)

**Figure 4.** Territorial distribution of the 8 clusters (Figure 4a–h; cluster sizes in brackets).

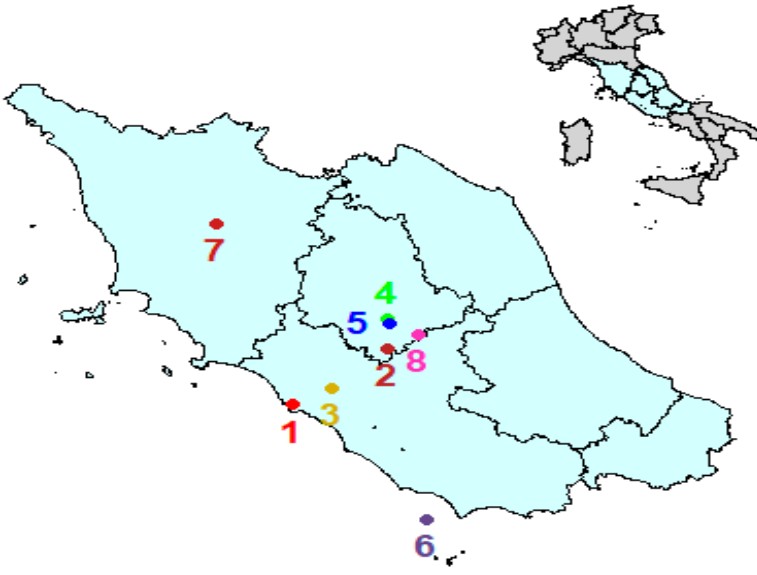

**Figure 5.** Geographic location of the mean centres of the 8 clusters.

### 3.3.2. Characterising Italian FFEFs

The final step of the research is the classification, both from the offer and the demand side, of the Italian farmhouses, which are grouped into the clusters shown in the figure above.

*Cluster 1: Small and Medium Farmhouses (SMFs) for Island Lovers*

Of the total 397 farmhouses investigated, this cluster has 69 (17.4%) units (Figure 3b); the average NB is equal to 19, so they are small and medium farmhouses (SMFs). Visitor evaluation of this cluster

was more concerned with the restaurants (foods and beverages), which had an average score of 9.23 (Table 6). The flow of activities and facilities in the surrounding areas offered by farmhouses grouped in this first cluster was considered to be quite good by guests, compared to the other clusters. However, the score on all farmhouse attributes is below the global mean (9.21) and the outdoor activities and facilities had a low score (8.74). The farmhouses of this group belong mainly to hilly Italian municipalities with peak concentrations in the two Italian islands: Sardinia (16%) and Sicily (14.5%). These islands, mainly characterised by a warm temperate climate, represent two important tourist areas in the Mediterranean Sea. However, the majority of the FFEFs are located near important beaches (e.g., Arzachena, Costa Smeralda, Cala Luna, Palau) or near important archaeological sites such as the Ispinigoli Caves, the Valley of Temples, and Syracuse. Furthermore, the main attractions and activities in the surrounding areas are the artistic, historical sites and nature reserves. The farmhouses of this group have an average NS above the global mean (8.35) and are more inclined to offer visitors activities and sports such as hiking, horse riding and fishing since their main characteristics are beautiful landscapes, gulfs (Orsei, Asinara) and lakes (Pergusa), as shown in Figure 4a.

*Cluster 2: Large Farmhouses (LFs) for Cultural Heritage Lovers*

In the second cluster, which consists of 26 farmhouses, the variables related to the dimensional attributes showed the highest average value for beds (53.2) and accommodation (17.5) compared to the global means, which are equal to 22.2 and 7.3 respectively, and when compared to other groups (Table 6). Furthermore, the most prominent characteristic of these farmhouses is that they offered, on average, the highest NS (10), above the global mean (7.63) and that of the other groups. All the FFEFs (100%) have connectivity (Wi-Fi). This may be due to the fact that 50% of FFEFs (Figure 4b) in this group are located in the municipalities of Umbria and Tuscany. It is well-known that these two regions are characterised by a long-standing tradition of agritourism facilities, certified quality food and tourist attractions, have a long and consolidated experience of offering overnight accommodation to large numbers of tourists and serve local traditional foods (especially oil and wine) produced directly on the premises. The SCOREACCOM (9.35) is also above the global mean (9.24) and so too is the SCORESETSUR (9.38 vs. 9.20). This could be a result of the fact that in these farmhouses, visitors can spend their holidays staying close to the most famous cities and the consolidated artistic and historical sites of cultural heritage such as Florence, Rome, Assisi, Spoleto, Gubbio, Siena, Todi, Orvieto, Volterra, San Gimignano and many others. This success is also due to the presence of FFEF websites that show, on average, a value above the global mean (4.22 vs. 3.85). As reported in Figure 4b, the remaining 50% of the farmhouses of this cluster are located in the municipalities of Lombardy (15.4%), Sicily (15.4%), Apulia (7.7%), Campania (3.9%), the Marche (3.9%) and Veneto (3.9%). These regions, like Umbria and Tuscany, have a significant endowment of nature reserves (Nebrodi Park, Alcantare Park, Etna Park, Cilento Park, Vallo Diano), of valleys (Trebbia, Staffora, Itria, and Taormina) and, above all, are surrounded by vineyards such as Franciacorta and Susegana. These farmhouses, located mainly in hilly and mountainous municipalities, offered a wide range of activities and facilities, with an average SCOREACTFAC (9.14) above the global mean (9.06), especially in reference to hiking, horse riding, shopping and tennis. In contrast, among the farmhouse attributes and visitor evaluation concerning the SCOREREST, there was an average value (9.38) below the global mean (9.43). Therefore, it can be argued that the evaluation of restaurants as a satisfaction factor for tourists staying in farmhouse accommodation is not as significant as satisfaction with attractions and amenities present in the surrounding areas.

*Cluster 3: Farmhouses with the Highest Consolidated Website Presence*

Belonging to this third group are 66 farmhouses (16.6%) with, on average, the highest consolidated presence of websites, which was comparable to the global mean (8.48 vs. 3.85) and the other groups. For this cluster, the websites have played an important role in determining visitor choice and evaluation.

The farmhouses of this cluster are situated mainly in municipalities in Tuscany in the Val d'Orcia (18.2%), in Liguria near the Magra-Montemarcello park (10.6%), in Sardinia, mainly in the province of Nuoro near the Gulf of Orosei (10.6%), in Sicily near the Lava Museum or the Butterfly House (9.1%) and in Umbria (7.6%). The remaining 44% are scattered across other Italian municipalities near important nature reserves (e.g., the Natural Park of Cilento in Campania, the Majella National Park in Molise, the *Ex cave di Cardona Nature* Park in Piedmont and Abruzzo's Nature parks), artistic and historical sites (e.g., the Acropolis of Temesa in Calabria, the International Museum of Ceramics in Emilia Romagna), museums and galleries, spas, theme parks and water parks (e.g., Liguria Acqui Terme, a spa city, where guests can enjoy discounted spa treatments with the Italian National Health Service and the Terme of Petriolo in Tuscany) and casinos in the Veneto region (Figure 4c). Another attraction is the presence of three casinos in the municipalities of Iesolo, Mascali and Olbia. For this cluster, the three variables related to the dimensional attributes (NB, NA, NBLA) and to the PNOG, which showed, on average, values below the global mean (Table 6). In contrast, for the variables related to the presence of Wi-Fi and other services, the average value is above the global mean (8.77 vs. 7.63). Regarding sport and activities in the surrounding areas, visitors can, above all, go hiking, horse riding, shopping and play tennis. As a consequence of these services offered by the farmhouses, the five variables considered in the analysis for the visitor evaluation highlight values above the global mean, especially in reference to the overall attributes (SCOREALL).

*Cluster 4: A Niche of Farmhouses Most Appreciated by Organic and Traditional Food Lovers*

Cluster 4 represents 19.6% of farmhouses considered (Figure 3b). These are luxurious agritourism operations, offering independent apartments, hotel-like treatment, with facilities such as horse clubs, spas, wellness centres, golf and so on. In the municipally of Tarquinia, in the province of Viterbo (Lazio), the farmhouse with the highest PNGO (€125) is located. This FFEF has a wellbeing centre with a swimming pool, whirlpool, sauna and a range of other wellness services. In the restaurant, guests can enjoy typical dishes mostly prepared using the farm's own produce and that of other local producers. Special menus are available for coeliacs and those with other food intolerances, coordinated by the restaurant manager, who is certified by the Italian Coeliacs Association. However, the prominent characteristic of the farmhouses regrouped in this cluster is the high price compared to the global mean (47.83 vs. 36.79) and to the other clusters. Furthermore, this group, compared to the other clusters, shows on average the best visitor judgments of the farmhouse attributes considered. It is possible to hypothesise that for the farmhouses of this group, the priority is to provide visitors with high-quality services, especially foods and beverages for the restaurant. Indeed, the group is characterised by the highest evaluation overall for the farmhouse attributes considered (9.82), especially for the restaurants (9.81). This cluster can be seen as a specific niche of farmhouses, 18% of which are located in the municipalities of Tuscany around Arezzo, Siena, Pisa and Massa Carrara; 14% are in the municipalities of the Apulian provinces (Lecce, Bari and Brindisi); and 11.5% are in provinces of Umbria (Perugia and Terni). The remaining 56.5% are scattered in other Italian municipalities located mainly near important tourist attractions such as ski resorts in Aosta city, the Pollino National Park, the Natisone Valleys, the Amalfi Coast and so on (Figure 4d). In these areas, organic and traditional foods are important factors for success. The variable (NR.NY) considered as a proxy for the presence of a farmhouse website highlights, on average, a value below the global mean (2.17 vs. 3.85). This suggests that a positive judgment made by visitors of the farmhouse attributes of this cluster is due to the reputation of these mature tourist areas and the foods and beverages that the agritourism operators offer, rather than to communication through websites. In line with a previous study, it is possible to hypothesise that the visitors of this cluster are more orientated towards tourist areas with a renowned reputation.

*Cluster 5: Family-Run Farmhouses (FRFs) for Authentic Food and Wine Lovers*

This cluster has the largest number of units (80 farmhouses, 20% of the total) with the smallest dimensions. Here, there is a presence of small traditional family-run farmhouses, characterised by small

rooms and a modest number of beds that do not always operate efficiently and offer a low NS (4.91 vs. a global mean of 6.75). However, in comparison with other clusters, farmhouses belonging to this group operate on a smaller scale in terms of the NB and the NA. The NB ranges from 2 to 35, with an average of 14.09 beds and the NA is between 1 and 11, with an average of 4.91 accommodation units. From a sustainability perspective, it is possible to hypothesise that the small dimensions could represent an advantage as they enable the preservation of cultural and product heritage and, that the attention of operators is focused more on the quality rather than quantity of services, foods and beverages offered. Indeed, this is appreciated by visitors who have made positive judgements about the farmhouse attributes with scores above the global mean, especially for the restaurants (9.75 vs. 9.43). The strong attractions of this cluster for visitors are represented mainly by the nature reserves (National Park of Gran Sasso and Monti della Laga, Majella National Park, Alta Murgia National Park, Cilento National Park); the countryside (Montalcino, Pienza, Collodi, Lucca, Florence, Pisa and so on); and the authentic food and wine (Montepulciano DOC and DOCG, Passerina and Pecorino wines). From the perspective of farmhouse websites that present both reviews and services, as reported in Table 6, the farmhouses of this group are less inclined, when compared to the global mean, to use websites to highlight their characteristics to visitors. From an analysis of Figure 4e, it emerges that 46% of farmhouses belong to Southern Italian municipalities, with a greater presence in Abruzzo (14%), Apulia (10%) and Campania (10%), while about 28% belong to the Northern municipalities of Italy and the remaining 26% of the municipalities are situated in the Centre of Italy, mainly in Tuscany (18.8%). In conclusion, the key success factors are a strong link with the territory, highlighted by the offering of one's own agricultural products and, human relationships, since in these farmhouses, the guests may have direct contact with the farmers and their families.

*Cluster 6: Medium and Large Farmhouses (MLFs) Less Appreciated by Visitors*

The great majority of farmhouses in this cluster (14 of 23, 61%) operate in low-lying areas (under 300 m ASL) and in particular, are situated in municipalities of Sicily (30%), Lazio (13%), Campania (8.7%) and Tuscany (8.7%), as shown in Figure 4f. The average size of the farmhouses in this cluster, expressed in terms of the NB and the NA, is 23.48 and 8.57 respectively; 56.5% of farmhouses have a NB below the global mean (22.16). From the analysis of farmhouse attributes, in contrast, the farmhouses belonging to this group highlight, on average, SCOREALL values below the global mean and lower values compared with the other clusters. However, the presence of a varied and significant natural and cultural heritage is not enough to make these areas attractive. Furthermore, the NS recorded, which is above the global mean (8.70 vs. 7.63), is not enough for visitor satisfaction. The services, like the other clusters, are represented mainly by Internet, air conditioning, a swimming pool, a playground for children, pet facilities, barbecues, a restaurant, educational opportunities and so on. Another critical weak point of this cluster, compared to the other clusters, is that the farmhouses, on average, have the lowest presence of websites (1.91) below the global mean (3.85), with the lowest number of written reviews given by visitors. This may be due to the fact that the farmhouses of this cluster are less visited when compared to the others groups.

*Cluster 7: Cheap Farmhouses for Undemanding Visitors*

The most prominent characteristics of this cluster, consisting of 23 farmhouses, are low Internet connectivity and low prices. In this group, 43% of farmhouses are without Wi-Fi and have the cheapest accommodation compared to the global mean (28.43 vs. 36.79) and to the other seven groups. Furthermore, this group of farmhouses offered visitors the lowest NS_Wi.Fi (5.39 vs. 7.63). The small dimensions of the farmhouses in terms of NA compared to the other clusters and to the global mean (3.83 vs. 7.30) and is also associated with minimal outdoor facilities (mainly parks for sporting activities). More than 26% of farmhouses are situated in low-lying and hilly municipalities of Liguria, 17.4% in the municipalities of Lombardy (mainly in the province of Como) and others are scattered in other Italian municipalities located near important tourist attractions such as the Gargano

National Parks, Valledoria (a well-known tourist site in the North of Sardinia), the Ombrone Valley in the Maremma Grossetana, the Umbrian hills between Lake Trasimeno and Perugia and Venice city centre (Figure 4g).

*Cluster 8: Large Farmhouses (LFs) with a Good Quality-Price Ratio of Attributes*

This cluster is characterised by farmhouses with, on average, an altitude of 299.26 ASL, which represent only 8% of the 397 farmhouses considered (Figure 3b). These large farmhouses with a NB and NA above the global means (36.28 vs. 22.16 and 10.34 vs. 7.30, respectively) belong mainly to municipalities of Tuscany (22%), with 37.5% equally distributed between Abruzzo, Apulia the Marche and Umbria. The remaining 40.5%, as shown in Figure 4h, are scattered in other Italian municipalities such as in Campania near the coasts of Cilento and near interesting locations such as Paestum, Pompei, Herculaneum, Amalfi, Positano, Salerno, Naples, and the volcano Vesuvius or near lakes such as Maggiore and Iseo in Lombardy. The PNGO is below the global mean (32.65 vs. 36.79) and the visitor evaluation referring to the SCOREALL, the SCOREACTFAC and the SCOREACCOM highlights values above the global means.

The visitors in these areas can have an active holiday with leisure, recreation and an opportunity to escape into nature. It is probable that in this cluster, like cluster 4, visitors chose their holiday destinations on the basis of the reputation of the most important tourist areas belonging to the municipalities where the farmhouses are located, and on the basis of recommendations from friends or colleagues, rather than on the basis of the presence of a website with reviews and services. However, the value of NR.NY is below the global mean (2.74 vs. 3.85) Table 6.

**Table 6.** Mean values of the starting variables in the 8 cluster solution, together with global means.

| Variables | Clus. 1 | Clus. 2 | Clus. 3 | Clus. 4 | Clus. 5 | Clus. 6 | Clus. 7 | Clus. 8 | Global Means |
|---|---|---|---|---|---|---|---|---|---|
| **ALTITUDE** | 268.45 | 403.69 | 300.02 | 346.04 | 350.9 | 306.09 | 183.00 | 299.56 | 314.15 |
| **Wi.Fi** | 0.93 | 1.00 | 0.97 | 0.92 | 0.76 | 0.96 | 0.57 | 0.94 | 0.89 |
| **NS** | 7.42 | 9.00 | 7.80 | 6.76 | 4.91 | 7.74 | 4.83 | 6.53 | 6.75 |
| **NB** | 19.29 | 53.19 | 21.65 | 18.69 | 14.09 | 23.48 | 16.04 | 36.28 | 22.16 |
| **NA.** | 7.09 | 17.46 | 6.68 | 7.12 | 4.29 | 8.57 | 3.83 | 10.34 | 7.30 |
| **NBLA** | 4.46 | 14.50 | 4.94 | 4.97 | 4.70 | 4.52 | 5.30 | 7.62 | 5.65 |
| **PNOG** | 37.7 | 39.89 | 34.03 | 47.83 | 30.29 | 37.85 | 28.43 | 32.65 | 36.79 |
| **SCOREALL** | 9.21 | 9.36 | 9.65 | 9.82 | 9.67 | 7.74 | 8.66 | 9.51 | 9.41 |
| **SCORESETSUR** | 8.90 | 9.38 | 9.42 | 9.69 | 9.36 | 7.77 | 8.62 | 9.13 | 9.20 |
| **SCOREACTFAC** | 8.74 | 9.14 | 9.33 | 9.64 | 9.23 | 7.38 | 8.15 | 9.16 | 9.06 |
| **SCOREACCOM** | 8.88 | 9.35 | 9.51 | 9.78 | 9.44 | 7.66 | 8.39 | 9.34 | 9.24 |
| **SCOREREST** | 9.23 | 9.38 | 9.63 | 9.81 | 9.75 | 7.65 | 8.93 | 9.43 | 9.43 |
| **NR.NY** | 3.46 | 4.22 | 8.48 | 2.17 | 3.12 | 1.91 | 3.10 | 2.74 | 3.85 |
| **NS_Wi.Fi** | 8.35 | 10.00 | 8.77 | 7.68 | 5.67 | 8.70 | 5.39 | 7.47 | 7.63 |

Source: Own work based on collected materials.

Finally, in relation to the last research question: How do different online visitor judgments contribute to the promoting of the natural and cultural resources present within different contexts of the Italian agritourism sector? The identification of the above "homogeneous" agritourism areas, each with their own distinct features, represents an important tool for understanding the different visitor satisfaction rates in different Italian agritourism contexts. This could have important implications for enriching the activities and the services offered by agritourism operators. Furthermore, improving existing activities and services and providing new ones as well as presenting farmhouses in an effective way through websites are crucial factors for farmhouse competition, above all for those operating in rural municipalities.

## 4. Discussion

This study has analysed the farmhouse sector in Italian municipalities from offer and demand perspectives in order to provide a more comprehensive framework for all stakeholders involved in the agritourism sector. Firstly, as Santeramo and Barbieri (2017) [47] suggest, it was necessary to highlight the different characteristics of FFEFs on the offer side in order to determine their suitability for the different types of visitor on the demand side. Indeed, the eight "homogenous" groups of FFEFs (Figure 4), identified through the application of HCA, combined the following aspects: farmhouse dimensions; prices; presence of a website that consolidates reviews and services; and visitor evaluations of principal farmhouse characteristics and attributes based on themes derived from the literature [64].

The results of the analysis indicated important factors that contribute to the promotion of the internal and external resources present in a "homogenous" agritourism context (e.g., Cluster 2). These included the physical dimensions of the FFEF, the distance from famous cities and sites of consolidated artistic and historic cultural heritage. This first factor influences both the process of productive diversification into agritourism activities and the efficiency of the FFEFs themselves. Indeed, as Barbieri and Mshenga (2008) [24] suggest, larger farms tend to be more successful as agritourism sites. In contrast, the distance from cities and heritage sites is an influential variable, as already highlighted by several authors [45–47] in attracting tourists and in generating tourist flows. Indeed, the present study clearly showed that the main attraction for visitors were the public access to the farm, the proximity to central cities and the quantity and quality of the natural resources (e.g., national and regional parks) together with archaeological and cultural heritage sites present in the areas where the FFEFs are located. These resources, in accordance with some studies [48,49], represent pillars in the development of tourism as a key economic sector. It is unquestionable that the agricultural landscape represents one of the most important drivers that guides customers in choosing a FFEF. For this reason, it is very important to promote and to communicate to visitors the undeniable value of this natural heritage. Indeed, the findings of the analysis, in line with previous studies [50–54], have highlighted that the key pull factors—most appreciated by visitors who had spent a period of their vacation in farmhouses present in almost of all the "homogenous" agritourism systems (clusters) identified, with the exception of Cluster 6—were the characteristics of the environment and landscape in the surrounding area.

Furthermore, the different agricultural landscapes positively affect the price of accommodation and drive tourists in their choice of a FFEF. This corroborates the views of some authors [64], who argue that agritourism is a series of types of services and products rather than one homogenous entity. Consequently, the findings of this study add a new contribution to the previous literature and offer readers a visual map (Figure 4) of the Italian farmhouse facility sector, a mosaic of natural landscapes where visitors can spend a period of their vacations on the basis of their preferences and their economic possibilities.

Finally, another important tool for promoting the high-quality landscape and services offered, especially in the case of FFEFs with small dimensions located in remote rural areas (e.g., Cluster 5), is the website. Several studies have shown that experienced tourists increasingly tend to book parts of their travels by themselves. Some studies have highlighted that, in the tourism sector, virtual communities can be used to enhance existing travel products and to create new divisions and capabilities [71]. In addition, other recent studies [35,72] have also mentioned that web pages need to offer more detailed and updated information to tourists. However, through the Internet, organisations, as well as individuals, can make their opinions, reactions and personal thoughts known to a global community of Internet users, so word of mouth (WOM) is being given new significance [73]. For this reason, it is very important that agritourism operators that operate in rural areas or that are less inclined to use a website understand the power of this tool to better enhance the characteristics and the attributes of their farmhouses and the uncontaminated natural capital present in the rural areas where they operate.

From the demand perspective, it seems that visitors, on average, were more satisfied with the restaurants (e.g., Cluster 4) than other farmhouse attributes. Consequently, in accordance with previous

studies [42,43], the local cuisine, which includes wine, is another important resource to be enhanced. Customer satisfaction is also influenced by price, something another study has confirmed. [74] (e.g., Cluster 8). Therefore, any attempt to measure satisfaction needs to take the value factor into account. Another study [75] suggests that the economical sustainability of resources is one of the most important aspects in the tourism business. Another resource that positively influences customer satisfaction is the nature of the human capital. Indeed, visitors who had spent a period of their vacations in the farmhouses of Cluster 5, for example, appreciated human relationships, since in these farmhouses, guests had had direct contact with the farmers and their families.

In light of the above, several contributions which deepen our knowledge of Italian FFEFs have been made. Firstly, where previous studies have broadly acknowledged the role of landscape and culinary tradition, this study has specifically demonstrated, through the investigation of visitor satisfaction with farmhouse attributes, the importance that visitor judgments can have in successful businesses within FFEFs. From this viewpoint, a broader application of this way of thinking may create many new opportunities for improving the services and the facilities of farmhouses, which have the potential to provide great social and economic benefit to farms, their locations and local communities. More specifically, the findings of the study show that the presence of local traditional foods produced directly on the premises and landscape attributes appear to play a significant role in agritourism consumer preferences.

Secondly, the paper provides a deeper understanding of the similarities/dissimilarities between the different groups of farmhouses spread across the Italian municipalities, and of how their different strategies can play a role in the development of the agritourism offer.

This research suggests there are great opportunities for developing new businesses in the farmhouse sector that are inextricably linked with some important resources: landscape, natural and cultural resources, restoration and heritage sites. From a managerial viewpoint, visitor judgments can positively affect the creation and positioning of new business models. This opportunity is particularly relevant given the growing demand for traditional and authentic experiences where historical, natural, cultural, and spiritual factors are key motivators for visitors. Another implication is the particular role that the FFEFs can play in the revitalisation of the community and surrounding municipalities. From a political perspective, FFEFs may become important catalysts for developing business models outside traditional spheres. An additional objective is to consider the wider implications of using the eight agritourism typologies as a tool to facilitate agritourism research. The outcome of this empirical research may enable planners, investors, destination managers and other stakeholders to better understand visitor expectations and formulate improved strategies, regional policy and a balanced approach towards educational agritourism development in Italian municipalities. Assessing visitor satisfaction and feedback can help managers to improve their service performance.

Last but not least, this study has clearly demonstrated that visitor satisfaction is a fundamental parameter, in accordance with Noe and Uysal (1997) [76], for the evaluation of the performance of destination activities, facilities, natural resources, services and products. Where once consumers trusted word of mouth (WOM) from friends and family, today they look to online comments (eWOM) for information about a product or service [77]. As a result, WOM can influence many receivers [78,79] and is viewed as a consumer-dominated marketing channel. Furthermore, in accordance with Jensen et al. (2014) [80], the years in business have a high influence on plans for the expansion of services.

## 5. Conclusions and Remarks

This paper provides three main contributions to agritourism literature:

(1) It presents a revised conceptualisation of agritourism, based on 397 farmhouses with educational farms incorporating the evaluations of 10,864 visitors;
(2) It represents one of the few studies that examines online reviews in the agritourism sector;
(3) It demonstrates how the different farmhouse typologies (clusters) can be used as a flexible research tool.

The study demonstrates a way to develop a tourist typology based on tourist interests, motivations, and activities with respect to the destination. The diversification of the offer is very important: most tourists come to the countryside to relax, but they also expect to be entertained in some way. The Principal Component Analysis acknowledges the importance of the farmhouse dimensions and visitor perceptions of farmhouse attributes for the designing of a specific policy and promotional measures to improve the supply of services in the farmhouse sector.

An important feature of the methodological approach was the focus on the dual perspectives of supply (dimension of accommodation, price, services, foods and beverages offered by farmhouse operators) and demand (visitor evaluations of principal farmhouse attributes). A better understanding of the similarities and differences between the judgments of those who use agritourism in Italian municipalities and the characteristics of the supply is an important outcome of this research and raises questions and has implications for practice. While the results of the research cannot be deemed representative of Italy as a whole, understanding differences and commonalities in evaluations within the eight groups of farmhouses may give suggestions for farmhouse management and wider agritourism policy. Another important outcome of the study is the possibility to implement more effective marketing strategies (e.g., emphasising opportunities for people to reconnect with the cultural roots of foods and to participate in typical activities and leisure activities on educational farms) and policy development (e.g., providing support for services and products that incorporate educational or local food components). Therefore, the identified farmhouses typology can be considered as a useful framework to inform and facilitate further research into farmhouse facilities with educational farms in Italian municipalities.

The study has some limitations that may also be viewed as opportunities for future research. Firstly, the data gathered are limited to one type of organisation—*agriturismi* with educational farms. It has been argued that farmhouses are a good setting to study the use of natural landscape resources for visitor satisfaction. Future research could further analyse positive and negative reviews given by visitors of the farmhouse attributes, with the aim of better understanding a range of problems in the agritourism sector. From a geographical perspective, this study has been confined to Italy, where the phenomenon of *agriturismi* with educational farms developed quite late when compared with other countries (in particular, Scandinavian countries such as Norway, Denmark and Sweden). While this was a logical starting point, similar studies in other countries with different landscapes and culinary cultures may produce different findings. In this regard, further study will ascertain the generalisability of the findings to different organisational and geographical contexts.

**Supplementary Materials:** The following are available online at http://www.mdpi.com/2071-1050/12/5/1749/s1, The dataset of the 11 variables employed in the study.

**Author Contributions:** Conceptualization, Analysis and Investigation by R.M.F.; Methodology by L.R.; Resources by R.M.F.; original draft preparation, writing, review and editing by R.M.F and L.R. All authors have read and agreed to the published version of the manuscript.

**Funding:** This research received no external funding.

**Conflicts of Interest:** The authors declare no conflict of interest.

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
