# Peer review of "Customer Satisfaction with Farmhouse Facilities and Its Implications for the Promotion of Agritourism Resources in Italian Municipalities"

_sustainability, doi:10.3390/su12051749_

Round 1
Reviewer 1 Report
I would like to congratulate to the Authors of that manuscript – it is a really good, interesting and mature piece of research. The theoretical framework grounds the problem under study through a complete and current range of references. The objectives are correctly defined. The methodology is well-designed and is consistent with the objectives of the study. The interpretation and discussion of results are clear, objective and consistent. The conclusions summarize well the results obtained and are consistent with the work presented. Below, you can find my minor remarks for further improving the manuscript.
In the Abstract and Introduction part, Authors should refer to how the topic is related to aspects of sustainability. This aspect is very well emphasized in the farther part of the manuscript, but due to the lack of it at the beginning, the reader can get the impression that it appears as if by accident.
In scientific writing, the use of "we" should be avoided (line 568).
I hope that Authors will continue their research. It would be really interesting to compare how the situation looks in other countries.
Author Response
Authors’ responses:
We would like to thank Reviewer 1 for the congratulations. We were very happy to read her/his comments.
In relation to the suggestions, we have explained both in the Abstract and in the Introduction (see from line 25 to line 29 and from line 70 to line 97) how the topic is related to the aspects of the sustainability. Furthermore, as suggested, we have replaced the personal form with the impersonal one (see lines 63, 738, 753, 874, 876, 879, 937, 939 and 944)

Reviewer 2 Report
The authors need to provide more information regarding the methods used that is:
- provide the data set and above all the tests done;
- provide information on the scale of the variables entered in the correlation matrix, PCA as well as CA;
- provide explanation on the soundness of using PCA and not hedonic regression as a previous step for CA.
Main criticism:
In contrast to CA, PCA is a multivariate statistical method that requires variables to be at least continuos and not as you report (erroneously) in table 2 discrete.
However I do think that you know it and that the scale reported in table 2 is just a typing mistake. If so please provide evidence for that.
Of course you can always reduce a continuous variable into a discrete variable whenever it makes sense (not here) but you loose information and, above all, you are not allowed anymore to perform some statistical tests such the ones that you did for instance correlation and PCA.
Once you give explanation on these methodological points I invite you to revise also the other minor points as follows:
- Introduction: Please cite seminal paper Sidali Schulze Spiller 2009 (ENTER, Springer) on the link between agritourism and online reviews
Table 1: Please provide Legend (FFEF) as you do it in the text
Author Response
Authors’ responses:
We wish to thank the Reviewer for her/his very useful comments and suggestions.
We numbered the different questions, in order to answer them point-by-point
- We will make available the dataset of the 11 variables employed in the study.
As to the tests, we did not make any, since we considered the data as a whole collective (exploratory point of view) and not as a sample from a wider population; we provide descriptive statistics to highlight the main features of the collective under study.
- Thank you for this observation; we made an addition to PCA method introduction, clearly stating that all the variables employed in PCA were the classic zero-mean unit-variance (quantitative) standardized ones. Of course, all of the remaining values (both in correlation matrices, both in CA) represent quantitative dimensionless variables, to allow for comparisons.
- Thank you for this request. We have realized to have written in an improper way the introduction to Section 2.2 (Data Analysis). As already said, we had no intention to employ models (such as, e.g., Quantile regression or Hedonic regression), but to perform a kind of analysis in which the variables enter in a symmetric way (we would like to cite one of the most preminent books of the French multivariate analysis school, Bouroche-Saporta p. 21: “la spécificité de l'analyse en composantes principales est qu'elle traite exclusivement de caractères numériques jouant tous le même rôle”).
In order to clarify our point of view, we added some lines at the beginning of the Section.
We also added two bibliographic references (you’ll find them at the end of this answer)
- We think there must have been a misunderstanding; what we intended is that the variables had to be quantitative (by “discrete”, we mean a quantitative variable assuming a (possibly infinite) discrete number of values, such as Binomial or Poisson in the case of random variables).
Anyway, in order to eliminate any possible source of misunderstanding, we deleted last column in Table 2, specifying in the text that all the variables are quantitative.
As for the tests regarding correlations, these latters are not to be intended in an inferential way, but only as (descriptive) measures of the association between couples of variables, without significance tests (Jolliffe, Cadima, pp. 2-4).
- We have cited (reference n.14), as suggested, the following paper: Sidali, K.L.; Schulze, H.; Spiller, A. The impact of online reviews on the choice of holiday accommodation. In: W. Höpken, U. Gretzel & R. Law (a cura di), Information and Communication Technologies in Tourism ENTER 2009.Vienna: Springer-Verlag, 87-98.
- We cited FFEFs as “farmhouse facilities with an educational farm” in full.
Added Refences (now in the paper, together with reference in point 5)
- 66. Bouroche J.M., Saporta G. L’analyse des données, Presses Universitaires de France. 1992
- 68. Jolliffe I.T., Cadima J. Principal component analysis: a review and recent developments. Trans. R. Soc. A 374: 2016, 20150202. http://dx.doi.org/10.1098/rsta.2015.0202

Reviewer 3 Report
Dear authors,
The paper analyses the very actual topic, discussing the customer satisfaction with farmhouse facilities and its implications for the promotion of agritourism resources in Italian municipalities. The authors notice that, the availability of online reviews referring to agritourism resources is very rarely addressed in academic literature, so this study investigates visitor satisfaction with farmhouses on the basis of online reviews available on the official websites of 397 farmhouses with educational farms associated with the Agritourism.it organization. Authors notice, that visitor satisfaction is important for successful destination marketing as it influences destination selection and the consumption of products and service. So authors point out, that this study’s findings provide insights for consumers and agritourism operators with the aim of matching customer expectations to the types of agritourism on offer in Italian municipalities
The paper is a good contribution to the agritourism discussion, it is clearly written and structured.
I agree with the authors, that „despite the growing body of online evaluation research in the tourism sector, in the agritourism sector the literature remains inconclusive regarding the evaluation of farmhouse attributes, especially when concerning the judgments made by visitors“, so the authors reached the goal to fill this gap and to contribute to a better understanding of the relationships between farmhouse features and visitor expressed satisfaction.
And I would like to share with authors the remark as well: the theoretical part of the article has some sections, but there seems missing deeper analysis of farmhouse facilities with educational farms from different countries experiences and scientific studies perspective, so it would be suggested to extend the section „Farmhouse Facilities with Educational Farms“.
Author Response
Authors’ responses:
We would like to thank Reviewer 3 for her/his useful comments and suggestions.
- We have explained (see from line 70 to line 97) in Section 1.2: Farmhouse Facilities with an Educational Farm, the way in which agritourism is related to the concept of sustainability.
- There are some authors’ perspectives of FFEFs and exhaustive definitions of Educational Farms provided by the Polish Ministry of Agriculture and Rural Development. (see from line 86 to line 91)

Round 2
Reviewer 3 Report
Thanks for the corrections